# Urban Inundation under Different Rainstorm Scenarios in Lin’an City, China

**DOI:** 10.3390/ijerph19127210

**Published:** 2022-06-12

**Authors:** Yan Chen, Hao Hou, Yao Li, Luoyang Wang, Jinjin Fan, Ben Wang, Tangao Hu

**Affiliations:** 1Institute of Remote Sensing and Geosciences, Hangzhou Normal University, Hangzhou 311121, China; chenyan_hznu@stu.hznu.edu.cn (Y.C.); houhao@hznu.edu.cn (H.H.); 2017210214023@stu.hznu.edu.cn (L.W.); 2019111008006@stu.hznu.edu.cn (J.F.); 20170056@hznu.edu.cn (B.W.); 2Zhejiang Provincial Key Laboratory of Urban Wetlands and Regional Change, Hangzhou 311121, China; 3Faculty of Geo-Information Science and Earth Observation (ITC), University of Twente, 7500 AE Enschede, The Netherlands; yao.li@utwente.nl

**Keywords:** urban inundation, InfoWorks ICM, designed rainfalls, Lin’an city

## Abstract

Under the circumstances of global warming and rapid urbanization, damage caused by urban inundation are becoming increasingly severe, attracting the attention of both researchers and governors. The accurate simulation of urban inundation is essential for the prevention of inundation hazards. In this study, a 1D pipe network and a 2D urban inundation coupling model constructed by InfoWorks ICM was used to simulate the inundation conditions in the typical urbanized area in the north of Lin’an. Two historical rainfall events in 2020 were utilized to verify the modeling results. The spatial–temporal variation and the causes of urban inundation under different designed rainfalls were studied. The results were as follows: (1) The constructed model had a good simulation accuracy, the Nash–Sutcliffe efficiency coefficient was higher than 0.82, R^2^ was higher than 0.87, and the relative error was ±20%. (2) The simulation results of different designed rainfall scenarios indicated that the maximum inundation depth and inundation extent increased with the increase in the return period, rainfall peak position coefficient, and rainfall duration. According to the analysis results, the urban inundation in Lin’an is mainly affected by topography, drainage network (spatial distribution and pipe diameter), and rainfall patterns. The results are supposed to provide technical support and a decision-making reference for the urban management department of Lin’an to design inundation prevention measures.

## 1. Introduction

The spatial and temporal distributions, intensities, and frequencies of rainfall are changing because of global warming [1]. Rainfall is becoming more uneven and intense, and these variations in rainfall have aggravated the occurrence of urban inundation, soil erosion, and other hazards [2,3]. Meanwhile, the urban expansion in China is accelerating, increasing the impervious surface of the city, changing the urban hydrological process and increasing the possibility of urban inundation [4,5,6]. The preferential flow process that occurs in cases of heavy rainstorms cannot be neglected, in which soil material and soil moisture have an important influence [7,8,9]. Urban inundation is a common issue that can occasionally become destructive and cause considerable losses in human life and economy [10,11]. For example, affected by the typhoon in July 2021, Yuyao, Zhoushan, and many other cities in China were flooded, affecting a total of 2.711 million people and causing direct economic losses of CNY 3.35 billion [12]. Therefore, in recent years, urban inundation has attracted increasing social attention [13,14]. The accurate simulation of urban inundation scenarios is important for the prevention of inundation hazards and the reduction in hazard losses.

Today, urban inundation models are widely used for studying urban inundation [15]. Scholars have developed many urban inundation models, including SSCM (Urban Storm Pipeline Calculation Model) [16], CSYJM (Urban Storm Runoff Model) [17], SWMM (Storm Water Management Model) [18], InfoWorks ICM (InfoWorks Integrated Catchment Management) [19,20], and MIKE series [21,22]. Models such as SSCM and CSYJM lack good pre- and post-processing programs, and most of them are based on applying research in a specific region [23]. These models are usually limited to the internal use of the development teams and are not widely used owing to a lack of promotion [24,25]. On the other hand, urban inundation models such as SWMM, the MIKE series, and InfoWorks ICM have a great number of users. Yu et al. constructed an urban inundation model of Jinan based on SWMM and used it to analyze the flood control measures of Jinan [26]. Patro et al. constructed a one-dimensional (1D) river model and a two-dimensional (2D) hydrodynamic model for the Delta region of the Mahanadi River Basin in India using MIKE 11 and MIKE 21, respectively [27]. Then they used MIKE FLOOD coupling to simulate the inundation extents and depths of the study area under different grades of flood. Sidek et al. used the InfoWorks ICM to construct a 1D river and 2D surface inundation coupling model in the Damansara catchment, Malaysia [28]. The model was verified by monitoring data, and the results showed that the model has a good accuracy (R^2^ > 0.8). These efforts by previous scholars proved the suitability of using inundation models to study urban inundation issues.

SWMM is one of the most widely used models because its code is open source [18]. However, as a 1D model, SWMM cannot suitably model surface flooding [29]. Commercial software such as the MIKE series models and InfoWorks ICM have good pre- and post-processing abilities to model surface flooding [30]. However, the MIKE series models (such as MIKE 11/MIKE 21) do not allow complete coupling. The MIKE FLOOD model is needed to couple the 1D model (MIKE 11/MIKE Urban) and the 2D model (MIKE 21) [31]. As opposed to the MIKE series models, InfoWorks ICM is the first model in the world to combine a 1D hydraulic model of urban drainage networks and channels with a 2D flood inundation model of the urban catchment in a single simulation engine [32]. Furthermore, the operation of InfoWorks ICM is easier than that of the MIKE series models, and the simulation accuracy is better than that of SWMM [25]. In this context, InfoWorks ICM is now widely used in urban flood management and assessment [33,34].

In this study, a hydrological model was constructed by InfoWorks ICM to detect the spatial patterns and affecting rules of urban inundation on the northern part of Lin’an City, Hangzhou. The main objectives of this study are: (1) to analyze the 2D surface inundation in the study area through the constructed fine hydrological model; (2) to analyze the spatial-temporal variation and the causes of urban inundation in the study area under designed rainfalls. This study is supposed to provide scientific support for making urban inundation prevention and control plans in Lin’an City.

## 2. Materials and Methods

### 2.1. Study Area

Lin’an City, located in the northwest of Zhejiang Province, China, is a county-level city governed by Hangzhou. The city experiences a mid-subtropical monsoon climate, which is warm and humid with abundant rainfall. This area is easily affected by monsoon and typhoon weather, with an average annual precipitation of 1613.9 mm. Continuous rainfall and short-term heavy rainfall occur frequently in this area. In recent years, Lin’an has experienced rainstorms many times, resulting in inundation in large parts of the urban area. These rainstorms have caused traffic jams and paralysis, which resulted in huge losses to people’s property and social economy. In June 2015, there was a sudden heavy rainfall in Lin’an, causing the collapse of 91 houses and a direct economic loss of CNY 180 million [35]. Similarly, in August 2019, a short-term rainstorm in Lin’an caused four deaths and a direct economic loss of CNY 169 million [36].

The selected study area is located in the north of the main urban area of Lin’an (30°14′55″ N–30°15′48″ N, 119°42′7″ E–119°43′27″ E) (Figure 1). It is a relatively independent catchment unit formed by the intersection of Agricultural and Forestry Road (AFR), Maxi River, Baini Road (BNR), and Shimen Ridge, with a vertical range of 31.34–135.14 m and an area of 1.89 km^2^. The study area is a rapid urbanization region, with construction land accounting for 87.83% of the total area. The population and residential buildings are relatively concentrated, and the roads (such as Maxi Road and Xishu Street) have been inundated many times, threating people’s lives and urban transportation. Therefore, it is a suitable site to simulate urban inundation and analyze its causes.

### 2.2. Data Collection and Pre-Processing

#### 2.2.1. Geographic Data

The geographic data applied in this study mainly included: (1) A digital elevation model (DEM) provided by the surveying and mapping department of Lin’an with a 2-m spatial resolution. The elevation range of the DEM is from 31.34 to 135.14 m. (2) A UAV image with 0.5 m spatial resolution, provided by the surveying and mapping department of Lin’an. The image was captured in December 2019. (3) Land-use data consisting of 21 land-use types (including residential land, industrial land, construction land, road, rivers, and other types). These data came from the surveying and mapping department of Lin’an. According to previous researches, the land-use types were classified into buildings, roads, green space, water, and others [25,37]. (4) Two historical rainfall events (29 May 2020 and 2 July 2020) were selected for modeling and derived from the meteorological department of Lin’an. (5) Drainage system data provided by the urban management department of Lin’an were used, which contained pipe diameter, pipe materials, and other information (Table 1).

#### 2.2.2. Designed Rainfall Events

In order to study the inundation in the study area under different return periods, rainfall peak position coefficients, and rainfall duration, a variety of rainfall scenarios were set up for the analysis. According to the Intensity–Duration–Frequency (IDF) formula of a Lin’an rainstorm proposed by the Zhejiang Provincial Department of Housing and Urban–Rural Development, the rainfall intensity of designed rainfall is calculated with the following formula [38]:(1)$$q=\frac{2763.132\times \left(1+0.399lgP\right)}{{\left(t+10.870\right)}^{0.753}}$$
where *q* is the designed rainfall intensity (L/(s·hm^2^)); *t* is the rainfall duration (min); and *P* is the return period (a). Rainfall scenarios of 60, 120, and 180 min and 1-, 2-, 5-, and 10-year return periods were used to simulate the urban inundation depths and extents. Related studies have shown that the rainfall peak position coefficient, r, in most areas ranges from 0.3 to 0.5 [39]. Thus, rainfall peak position coefficients were set to 0.3, 0.4, and 0.5. In this study, a total of 36 rainfall scenarios were designed for analysis. Figure 2 presents a section of the hydrograph of the designed rainfall events (taking the rainfall duration of 120 min as an example).

### 2.3. Overall Workflow

This study included three main parts: the construction of InfoWorks ICM model, model validation, and simulation of different designed rainfalls (Figure 3). First, we input the processed geographic data (DEM data, drainage system data, land-use data, and historical rainfalls) into the InfoWorks ICM model. Then, the model was verified by the observed flow data and investigation results of inundation depths. Finally, we used the constructed model to simulate inundation depths and extents under different designed rainfall scenarios. According to the results, the spatial-temporal variation and causes of urban inundation under historical rainfalls and designed rainfalls were analyzed.

### 2.4. Model Description

As one of the best urban inundation models, InfoWorks ICM can achieve integrated 1D and 2D hydrological–hydraulic modeling. The model has many advantages. For example, it can shorten the modeling time with its parallel computation based on GPUs, especially 2D hydrodynamic modeling [25]. In addition, data in ArcGIS, AutoCAD, and Excel formats can be imported into InfoWorks ICM, which makes data processing easier [40]. Furthermore, the model has many modules, including hydrological, 1D pipe network flow hydraulic, 2D urban inundation, and real-time control modules [41]. The main modules involved in this study were the hydrological, 1D pipe network flow hydraulic, and 2D urban inundation modules. The hydrological module serves to calculate urban surface rainfall runoff according to the spatial division of the sub-catchment area and the surface composition of different runoff characteristics. Its main calculation modules include a runoff model and a routine model. In the hydraulic calculation of 1D pipe network, the flow velocity and water depth of the pipeline are simulated by solving the Saint-Venant equations (Equations (2) and (3)). The 2D urban inundation process is implemented by solving shallow water equations with the 2D finite volume method (Equations (4)–(6)):(2)$$\frac{\partial A}{\partial t}+\frac{\partial Q}{\partial x}=0,$$
(3)$$\frac{\partial Q}{\partial t}+\frac{\partial }{\partial x}\left(\frac{Q^{2}}{A}\right)+gA\left(\cos\theta \frac{\partial h}{\partial x}-S_{0}+\frac{Q\left|Q\right|}{K^{2}}\right)=0,$$
(4)$$\frac{\partial h}{\partial t}+\frac{\partial \left(hu\right)}{\partial x}+\frac{\partial \left(hv\right)}{\partial y}=q_{1D},$$
(5)$$\frac{\partial \left(hu\right)}{\partial t}+\frac{\partial }{\partial x}\left(hu^{2}+\frac{gh^{2}}{2}\right)+\frac{\partial \left(huv\right)}{\partial y}=S_{0,x}-S_{f,x}+q_{1D}u_{1D},$$
(6)$$\frac{\partial \left(hv\right)}{\partial t}+\frac{\partial }{\partial y}\left(hv^{2}+\frac{gh^{2}}{2}\right)+\frac{\partial \left(huv\right)}{\partial x}=S_{0,y}-S_{f,y}+q_{1D}v_{1D}$$
where *Q* is the flow (m^3^/s); *A* is the cross-sectional area of the pipeline (m^2^); *t* is the time (s); *x* is the length along the *x* direction (m); *h* is the water depth (m); *g* is the gravity acceleration (m/s^2^); *θ* is the horizontal angle (degree); *K* is the conveyance and is calculated using the Colebrook–White formula or the Manning formula; $S_{0}$ is the bed slope; *u* and *v* are the velocity components in the *x* and *y* directions (m/s), respectively; $S_{0,x}$ and $S_{0,y}$ are the bed slope components in the *x* and *y* direction (m^2^/s^2^), respectively; $S_{f,x}$ and $S_{f,y}$ are the friction components in the *x* and *y* directions (m^2^/s^2^), respectively; $q_{1D}$ is the outflow per unit area (m^3^/s); $u_{1D}$ and $v_{1D}$ are the velocity components of $q_{1D}$ in the *x* and *y* directions (m/s), respectively.

### 2.5. Model Setup

#### 2.5.1. Setup of Hydrological Model

The process of constructing the hydrological model mainly included: (1) the division of the sub-catchment area and (2) the determination of the runoff model and routine model. First, the sub-catchment areas were divided by the Thiessen polygon method based on the distribution of nodes. Then, the sub-catchment areas were modified according to the wall structure, building, slope, and aspect. The nearby sub-catchment without nodes flowed into the adjacent sub-catchment according to the aspect. In this study, a total of 247 sub-catchment areas were obtained. The nature of the underlying surface can affect the processes of rain water infiltration and confluence, so it was necessary to extract the runoff surface of the sub-catchment according to the land-use types. The Area Take Off (ATO) tool in InfoWorks ICM was used to extract the runoff surface type and its proportion in each sub-catchment. The parameters of each runoff-producing surface are shown in Table 2.

#### 2.5.2. Setup of 1D Pipe Network Hydraulic Model

The construction of the 1D pipe network hydraulic model involved the construction of the drainage system, which was needed to deal with the original drainage system data. The main process was deletion of the rain grate and its connecting pipeline. Because the rain grate was connected to the drainage system through a thin and short pipeline, its deletion had little impact on the final simulation results. Finally, there were 245 pipelines, 244 manholes, and 5 outfalls in the study area, and the pipe diameter ranged between DN200 and DN2000. The Manning coefficient of the pipeline is an important input parameter of InfoWorks ICM, and different materials have different values. The pipelines in the study area were mainly made of concrete and plastic. Referring to the SWMM user manual [42], it was determined that the Manning coefficient of the concrete pipeline was 0.013, and that of the plastic pipeline was 0.012.

#### 2.5.3. 1D–2D Model Coupling

In InfoWorks ICM, the 1D model was mainly used to simulate the drainage network flow, and determine the overflow nodes and overflow volume. The 2D model was used to simulate the diffusion of overflow generated by the drainage network on the complex surface. Through this process, the inundation depths and extents were obtained. The first step was building a 2D urban inundation model for the ground elevation model. The main process was extracting the elevation points from the DEM data, and then importing them into InfoWorks ICM to establish the TIN (Triangulated Irregular Network) model. Then, the 2D interval mesh was divided in the study area. The minimum mesh resolution was 25 m^2^, and the maximum was 100 m^2^. In this process, buildings were regarded as non-flooding zones and were not meshed [43]. In addition, considering that the road is generally lower than the sidewalks on both sides, its elevation was reduced by 15 cm and meshed. Finally, we set the flood type of manholes as “2D”, which achieved 1D and 2D model coupling.

### 2.6. Model Validation

In order to ensure the accuracy and applicability of the model, the observed data were used to verify the model. The two historical rainfalls on 29 May 2020 and 2 July 2020 were selected as the rainfall input data. The model was verified by comparing the observed discharge of flow monitoring points and the simulation results of the model. The coefficient of determination (R^2^), root mean square error (RMSE), and Nash–Sutcliffe efficiency coefficient (*NSE*) were used to evaluate the simulation results. The formula of NSE is as follows:(7)$$NSE=1-\frac{\sum_{t=1}^{T}{\left(Q_{o}^{t}-Q_{m}^{t}\right)}^{2}}{\sum_{t=1}^{T}{\left(Q_{o}^{t}-{\bar{Q}}_{o}\right)}^{2}}$$
where $Q_{o}^{t}$ is the observed flow at *t* time (m^3^/s), $Q_{m}^{t}$ is the simulated flow at t time (m^3^/s), and ${\bar{Q}}_{o}$ is the average value of observed flow (m^3^/s).

## 3. Results

### 3.1. Simulation Results of Historical Rainfalls

Figure 4 shows the validation results of the two historical rainfalls. Table 3 summarizes the simulated and observed flows in the validation processes. The NSE of the two historical rainfalls was more than 0.82, the R^2^ was above 0.87, and the relative error was ±20%, as shown in Table 3. In addition, after investigating the inundation depths and extents during the historical rainfalls in the study area, some areas were detected in inundation. Table 4 presents the recorded and simulated maximum inundation depths during the 2 July 2020 rainfall event. The table shows that the simulated maximum depth of the model was similar to the recorded depth. After calculation, the relative errors of MXR, XSS, and SLS were 15.5%, –17%, and –13%, respectively. Although there were some differences between the simulation results and the recorded results, the differences were in an acceptable range. The constructed model was able to simulate the inundation process in Lin’an and could be used for the following studies.

### 3.2. Simulation Results of Designed Rainfalls

#### 3.2.1. Different Return Periods

The designed rainfalls of different return periods (1-, 2-, 5-, and 10-year), different rainfall peak position coefficients (r = 0.3/0.4/0.5), and different rainfall durations (*t* = 60 min/120 min/180 min) were input into the constructed model for simulation. After simulation, statistical analyses of the inundation depths and extents were accomplished. Statistics included the number of overflow nodes, maximum overflow volume of nodes, maximum inundation depth, and inundation extent under different designed rainfall scenarios. Table 5 and Figure 5 present the inundation depths and extents of the study area under different designed rainfall scenarios (*t* = 120 min).

Figure 5 shows the spatial distribution of inundation under different designed rainfall scenarios. The results indicated that the inundation of the study area became more serious as the return periods increased, and the inundation extents in the south of the study area were increasing. Table 5 presents the inundation depths and extents under different designed rainfall scenarios. When the rainfall peak position coefficient was 0.4, the number of overflow nodes was 13, 18, 31, and 37, respectively, in the return periods of 1, 2, 5, and 10 years. Similarly, the maximum overflow volumes of the nodes were 0.5 m^3^/s, 0.6 m^3^/s, 0.8 m^3^/s, and 0.9 m^3^/s, respectively. The maximum inundation depths were 0.847 m, 0.978 m, 1.126 m, and 1.222 m, respectively. In correspondence, the inundation extents were 84,453 m^2^, 103,769 m^2^, 148,234 m^2^, and 195,118 m^2^. Thus, with the same rainfall peak position coefficient, the number of overflow nodes, maximum overflow volumes of nodes, maximum inundation depths, and inundation extents increased as the return periods increased. Furthermore, through comparing the contribution areas of inundation depths under different return periods, the area with a depth of less than 0.2 m increased the most. The statistical results showed that the inundation area with a depth of less than 0.2 m accounted for about 90% of the increased area from 1 year to 10 years.

#### 3.2.2. Different Rainfall Peak Position Coefficients

Table 5 shows that when the return period was 5 years, the numbers of overflow nodes under different rainfall peak position coefficients were 28, 31, and 32, respectively. The maximum overflow volumes of the nodes were 0.7 m^3^/s, 0.8 m^3^/s, and 0.8 m^3^/s. The maximum inundation depths were 1.096 m, 1.126 m, and 1.139 m, and the inundation extents were 137,329 m^2^, 148,234 m^2^, and 158,139 m^2^. Different return periods had similar results. The results showed that the number of overflow nodes, the maximum overflow volume of nodes, maximum inundation depths, and inundation extents increased with the increase in rainfall peak position coefficients in the same return period. By comparing the inundation conditions under different scenarios (see Table 5), we found that the inundation area with a depth of less than 0.8 m increased gradually, while the area with a depth of over 0.8 m remained almost unchanged. The lag of the rainfall peak increased the inundation area, and 99% of the increased area came from the area with a depth of less than 0.8 m. The area with a depth of less than 0.2 m increased most, accounting for 87% of the increased area (Figure 6).

#### 3.2.3. Different Rainfall Durations

Table 6 presents the inundation depths and extents under different rainfall durations when the rainfall peak position coefficient was 0.4. With the increase in rainfall duration, the cumulative rainfall, maximum inundation depth, and inundation extents all increased in the same return period. However, the maximum inundation depth increased little as the rainfall duration increased. When the rainfall duration increased from 60 to 180 min, the maximum inundation depth increased by 0.06 m on average. Compared with the maximum inundation depth, the cumulative rainfall and inundation extent changed more significantly. Taking the 10-year rainfall events as an example, when the rainfall duration increased from 60 to 180 min, the cumulative rainfall increased by 42.20%, from 56.4 to 80.2 mm. Meanwhile the inundation area increasing by 10.92%, from 181,229 to 201,027 m^2^. These results indicated that the rainstorm with a longer rainfall duration brought more rainfall, which resulted in a larger inundation extent (Figure 7).

## 4. Discussion

### 4.1. Urban Inundation under Historical Rainfall

In this study, an InfoWorks ICM model was employed to build a 1D pipe network and 2D urban inundation coupling model. The values of the NSE, R^2^, and RMSE in the validation results of the coupled model were similar to the results of Zhang et al. [41] and Liew et al. [44]. This showed that the 1D–2D coupled model constructed in this study was reliable and able to accurately simulate the inundation depths and extents in the study area. Figure 8 shows that in the rainfall event on 2 July 2020, there were three obvious inundation areas in the study area, namely, MXR, XSS, and SLS. MXR was the most serious inundation area, with a maximum inundation depth of 20 cm. MXR is located in the west of the study area, whose topography in the south is about 80 cm higher than that in the north. Moreover, there is almost no pipeline distribution and only one manhole in the south. During the rainstorm, this manhole could not be discharged in time after gathering too much rainwater and thus caused overflowing. Rainwater flowed from south to north to form an inundation in low-lying areas, according to the topography (Figure 9). As a result, the characteristics of inundation in this area were deep in the north and shallow in the south, and the inundation area was wide. The inundation analysis results showed that the two important factors affecting the formation of inundation were topography and pipe network distribution. In order to avoid serious inundation in MXR, the low-lying areas should be filled. Moreover, additional drainage outlets should be added to speed up the discharge of rainwater. The simulation results also showed that the InfoWorks ICM model could simulate inundation depths and extents according to the local micro-topography of the study area.

### 4.2. Urban Inundation Projections under Different Designed Rainfalls

The inundations in the study area showed a similar pattern under different designed rainfall scenarios. With an increase in the return period, rainfall peak position coefficient, and rainfall duration, the maximum inundation depth and inundation extent increased. As a result, the urban inundation became more serious. This result was consistent with previous studies of urban areas that considered the influence of different designed rainfalls on urban inundation [45,46,47]. From the perspective of spatial distribution, the inundation areas are mainly concentrated in the south of the study area (Figure 10) because the topography in the south of the study area is lower than that in the north. During the rainstorm, most rainwater flowed southward and into the drainage network in the south. Compared with the drainage network in the north, the drainage network in the south was under higher drainage pressure. Therefore, the southern areas were more likely to be inundated than the northern ones. This was similar to the spatial distributions of inundation found in the Chebei Watershed [48], which were mainly concentrated in the lower terrain areas.

In order to further analyze the change in inundation of the study area, four areas (Figure 10A–D) were selected for detailed analysis. XJSR (Figure 10A) is located in the middle of the study area with an elevation ranging from 36.4 m to 38.8 m. The diameter of the drainage network in this area is generally 2000 mm. Under the designed rainfall scenarios, there was no inundation in this area. This result showed that the areas with higher terrain and larger pipe diameter are not easily inundated. NLS (Figure 10B) is located in the northwest of the study area and is higher than XJSR. In contrast to XJSR, the diameter of the drainage network in NLS is 600 mm. This area was inundated under the 1-year rainfall scenario. With the increase in return periods, the inundation depth increased. Under the 10-year rainfall scenario, the inundation depth in some areas was over 0.6 m. This happened because, during the rainstorm, the drainage network could not accommodate lots of rainwater and thus overflowed. Although some rainwater flowed to the lower places under the influence of the terrain, most of the rainwater still flowed into the drainage network. This result indicated that areas with higher terrain and smaller pipe diameter can cause inundation. AFR (Figure 10C) is located in the south of the study area and the terrain is flat (the elevation range is 35.3–35.6 m). The diameters of most of the pipes in the drainage network are 1500 and 1800 mm. This area began to be inundated under the 5-year rainfall scenario. The maximum inundation depth was 0.104 m under the 10-year rainfall scenario. Compared with XJSR, the result of the AFR showed that in the case of a similar pipe diameter, inundation was more likely to occur in lower terrain areas. In addition, AFR was inundated only under the 5-year rainfall scenario, which indicated that rainfall pattern was an important factor affecting the inundation. WSS (Figure 10D) is located in the south of the study area, and the terrain is flat. The diameter of the drainage network is small, only 300 mm. Thus, the drainage network could not accommodate much rainwater under the designed rainstorms, forming inundation in the surrounding area. The maximum inundation depth was 0.442 m here. Compared with AFR, the result showed that areas with lower terrain and smaller pipe diameter will be seriously inundated. Overall, urban inundation is mainly affected by topography. However, from a local point of view, urban inundation is not only affected by topography but also affected by the drainage network and rainfall patterns. In summary, inundation not only occurs in areas with low terrain but also occurs in areas where the pipe diameter is too narrow. Moreover, a short-term rainstorm is more likely to inundate cities. Li et al. [49] and Cheng et al. [50] reached a similar conclusion in their study of urban inundation. However, compared with them, we clearly and intuitively showed the impact of different factors on inundation through the presentation of data and details. For the urban management department of Lin’an City, detailed modifications including increasing are small pipe diameters and filling low-lying areas is recommended. Moreover, LID (Low-impact development) techniques, such as permeable pavements, green roofs, and rain barrels, can be implement in the study area to reduce overflows. In addition, we suggest installing flow control devices (gates) in overflowing manholes to reduce inundation and arranging drainage pumps in inundated areas to facilitate the timely removal of accumulated water. Now, we only propose ways to reduce overflows, but how to implement it needs further research with InfoWorks ICM because it could simulate pipe diameter expansion scenarios and different LID scenarios.

### 4.3. Limitations

Several limitations need to be addressed for a comprehensive understanding in this study. First, the parameters of the 1D–2D coupled model are not universal and only applicable to the current study area. The parameters must be adjusted when the model is extended to other study areas. Parameters can be selected by consulting the InfoWorks ICM user manual and the SWMM user manual. Second, the influence of the rivers was not considered in the modeling process. When there is a serious rainstorm, the excessive water level of the rivers will affect the discharge of rainwater and then affect the inundation depths and extents. Therefore, it is be necessary to collect information such as riverbed elevation and river cross-sectional coordinates to construct the river model in future studies, to improve the accuracy of the model. Third, the land-use data in this study were extracted from historical remote sensing images, which do not reflect the latest land use in the study area and may create uncertainty in the model. Finally, soil moisture will affect the simulation results. The widely used Horton’s model is used in the model construction. However, the model does not account for initial moisture conditions adequately [51]. In future studies, we can try to use the Green–Ampt infiltration model or other runoff models for simulation and compare the simulation accuracy.

## 5. Conclusions

In this study, a 1D–2D coupled model constructed with InfoWorks ICM was used to simulate the inundation scene in a typical inundation area in the north of Lin’an’s main urban area. The model was verified by two historical rainfalls. Different designed rainfalls were input into the coupled model to analyze the spatial-temporal variation in urban inundation in the study area. The main conclusions are listed below:(1)The simulation results of the 1D–2D coupled model were acceptable. The validation results of two historical rainfalls showed that the NSE values were all above 0.82, the R^2^ values were all above 0.87, and the relative errors were all ±20%. Meanwhile, the simulating depth was consistent with the observed depth. These results indicated that the coupled model could accurately simulate the inundation situation in the study area.(2)Scenario analysis demonstrated that with the increase in the return period, rainfall peak position coefficient, and rainfall duration, the maximum inundation depth and inundation extent increased. The increased inundation area was mainly concentrated in the south of the study area. Combined with the change in the inundation of the local area, inundation mainly occurred in areas with low terrain (such as AFR), lack of drainage network (such as MXR), and narrow pipe diameter (such as WSS). Rainfall patterns also affected inundation, as in the case of AFR.

The inundation analysis under the designed rainfall scenarios will help us better understand the main causes and changing characteristics of inundation in Lin’an. Furthermore, it will help policy makers in Lin’an City to formulate appropriate urban inundation prevention measures, such as increasing small pipe diameters or filling low-lying areas, to reduce the risk of urban inundation. The results of this study can also be used as a reference for solving urban inundation problems in other cities with a similar geological and climatic condition.

## Figures and Tables

**Figure 1 ijerph-19-07210-f001:**
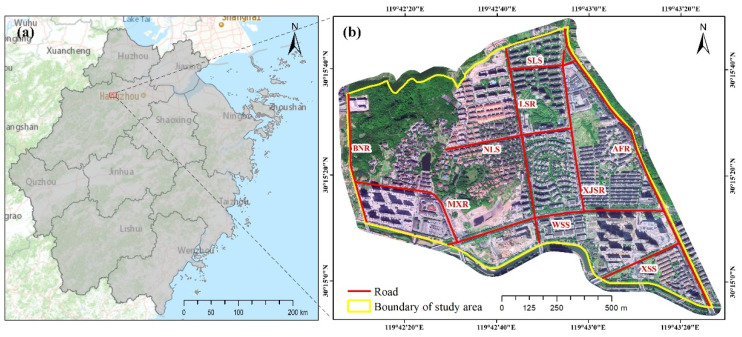
Location of the study area. AFR, Agricultural and Forestry Road; SLS, Shuanglin Street; LSR, Linshui Road; NLS, Nonglin Street; WSS, Wusu Street; XSS, Xishu Street; MXR, Maxi Road; XJSR, Xijingshan Road; BNR, Baini Road. (**a**) Zhejiang Province (**b**) Study area.

**Figure 2 ijerph-19-07210-f002:**
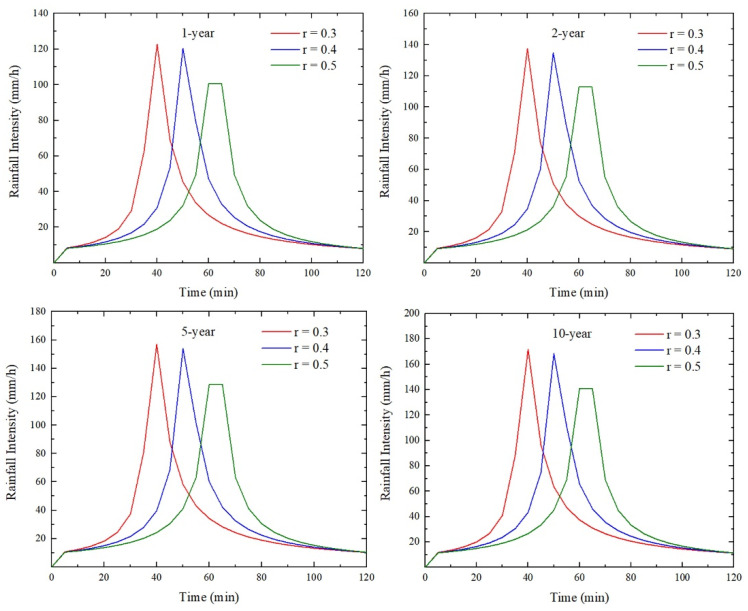
Hydrographs of the designed rainfall events (*t* = 120 min).

**Figure 3 ijerph-19-07210-f003:**
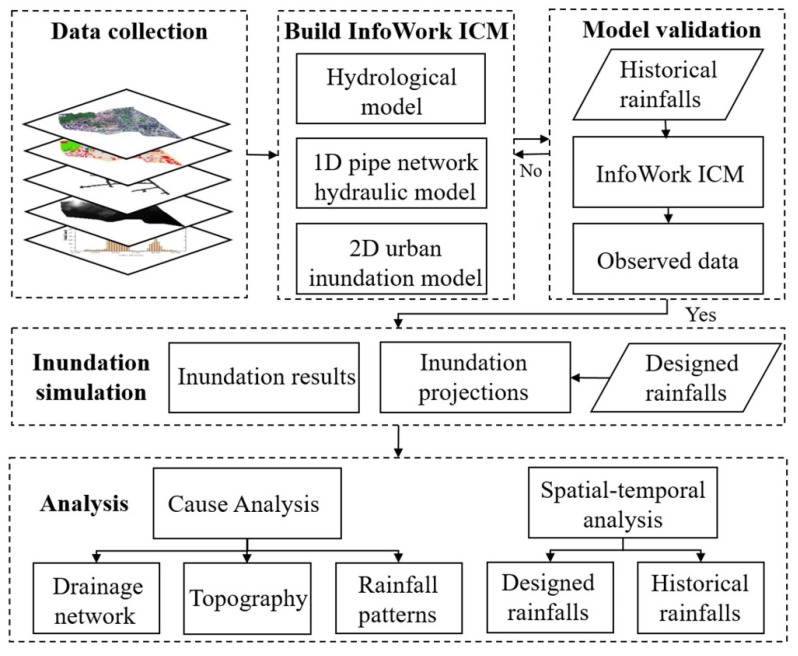
Technical flowchart.

**Figure 4 ijerph-19-07210-f004:**
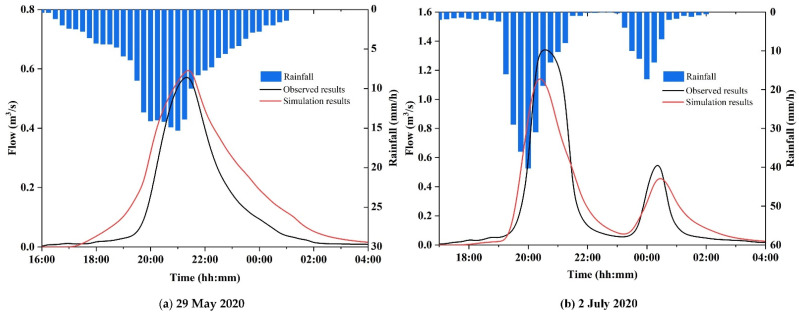
Comparison between simulated and observed flows.

**Figure 5 ijerph-19-07210-f005:**
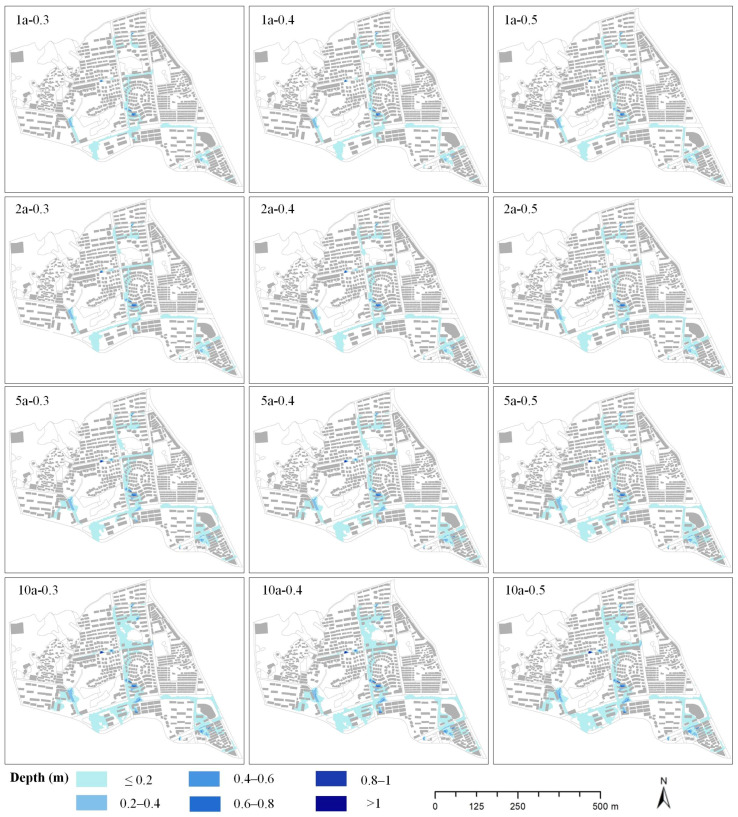
Simulation results under designed rainfall scenarios, where “1a-0.3” means *P* = 1a and r = 0.3 (*t* = 120 min).

**Figure 6 ijerph-19-07210-f006:**
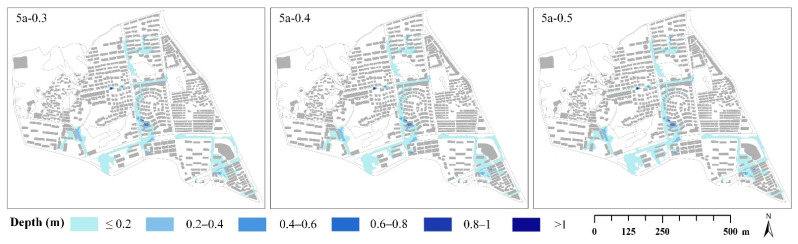
The inundation scene with different rainfall peak position coefficients (*t* = 120 min and *P* = 5a).

**Figure 7 ijerph-19-07210-f007:**
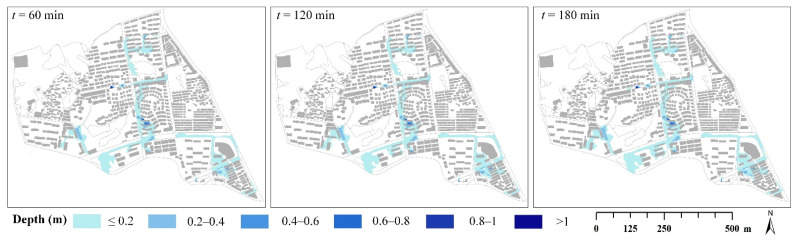
The inundation scene at different durations (*P* = 5a and r = 0.4).

**Figure 8 ijerph-19-07210-f008:**
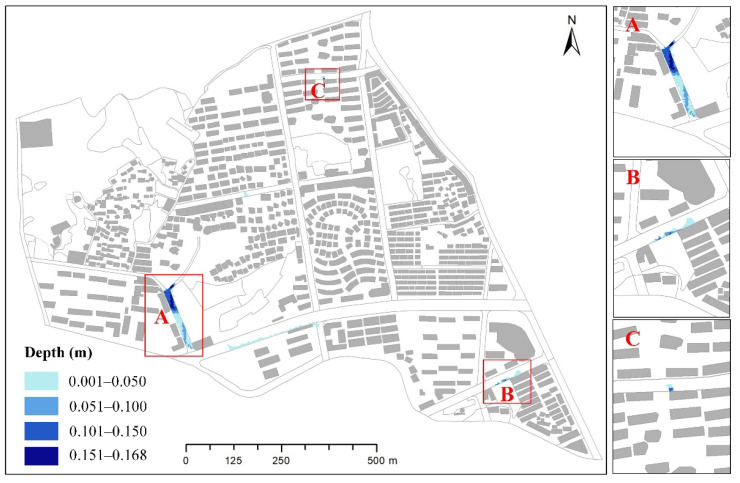
The inundation result from rainfall on 2 July 2020 ((**A**) is MXR, (**B**) is XSS, and (**C**) is SLS).

**Figure 9 ijerph-19-07210-f009:**
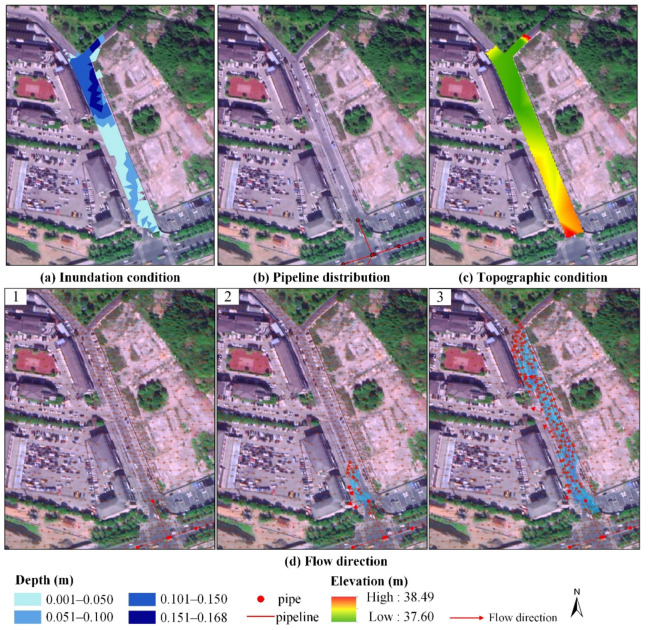
The inundation situation in MXR (the image came from aerial photography).

**Figure 10 ijerph-19-07210-f010:**
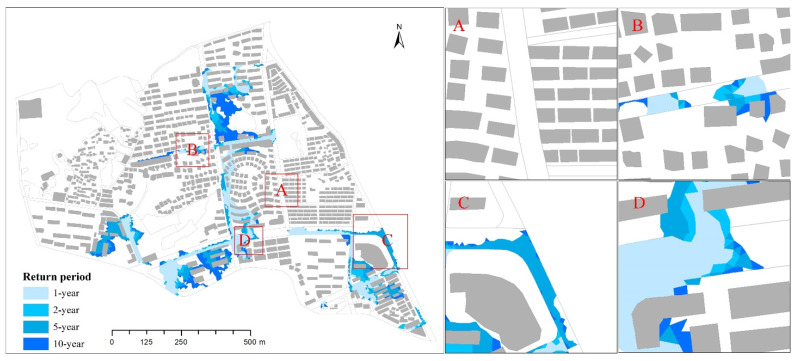
Inundation of the study area during different return periods (*t* = 120 min and r = 0.4): (**A**) XJSR, (**B**) NLS, (**C**) AFR, (**D**) WSS.

**Table 1 ijerph-19-07210-t001:** List of geographic data in the study area.

Type	Format	Resolution (m)	Main Attributes	Data Source
DEM	GeoTIFF	2	Elevation	Surveying and mapping department of Lin’an
UAV image	GeoTIFF	0.5	—	
Land-use data	Shapefile	—	Land-use type	Meteorological department of Lin’an
Rainfall data	Excel	—	Time and rainfall	
Drainage system data	Shapefile	—	Pipe diameter, pipe materials	Urban management department of Lin’an

**Table 2 ijerph-19-07210-t002:** Parameter attributes for five kinds of surfaces.

Surface Type	Routine Model	Routine Parameter	Surface Type	Runoff Model	Runoff Coefficient	Initial Loss (m)
Road	SWMM	0.02	Impervious	Fixed	0.9	0.0015
Building	SWMM	0.02	Impervious	Fixed	0.8	0.001
Others	SWMM	0.025	Impervious	Fixed	0.5	0.005
Water	SWMM	0.03	Impervious	Fixed	1	0
Green space	SWMM	0.2	Pervious	Horton	–	0.005

**Table 3 ijerph-19-07210-t003:** Model validation results.

Events	Peak Flow (m^3^/s)		NSE	R^2^	RMSE	Relative Error (%)
	Record	Simulation				
29 May 2020	0.569	0.603	0.82	0.94	0.07	6.0
2 July 2020	1.351	1.150	0.85	0.87	0.15	–14.9

**Table 4 ijerph-19-07210-t004:** Statistics of the maximum inundation depth of the 2 July 2020 rainfall event.

Number	Position	Recorded Depth (cm)	Simulated Depth (cm)	Errors (cm)
1	MXR	20	16.9	3.1
2	XSS	10	11.7	–1.7
3	SLS	10	11.3	–1.3

**Table 5 ijerph-19-07210-t005:** The inundation depths and extents in the study area under different designed rainfall scenarios (*t* = 120 min).

Return Period (a)	r	Number of Overflow Nodes	Maximum Overflow Volume of Nodes (m^3^/s)	Maximum Inundation Depth (m)	Inundation Extent (m^2^)	Contribution Area of Different Inundation Depths (m^2^)					
						≤0.2 m	0.2–0.4 m	0.4–0.6 m	0.6–0.8 m	0.8–1 m	>1 m
1	0.3	12	0.5	0.813	80,781	75,772	3515	1161	309	24	0
	0.4	13	0.5	0.847	84,453	78,710	4228	1115	376	24	0
	0.5	13	0.5	0.86	87,483	81,457	4511	1115	376	24	0
2	0.3	16	0.6	0.946	100,981	93,347	5809	1190	564	71	0
	0.4	18	0.6	0.978	103,769	95,645	6299	1120	620	85	0
	0.5	20	0.7	0.99	106,932	98,468	6603	1156	620	85	0
5	0.3	28	0.7	1.096	137,329	126,259	8708	1147	710	157	48
	0.4	31	0.8	1.126	148,234	136,652	8957	1710	710	134	71
	0.5	32	0.8	1.139	158,139	146,146	9368	1579	841	134	71
10	0.3	36	0.8	1.203	188,477	173,377	11,171	2702	920	222	85
	0.4	37	0.9	1.222	195,118	178,884	11,843	3084	1000	222	85
	0.5	44	0.9	1.235	201,744	185,235	11,974	3218	1010	155	152

**Table 6 ijerph-19-07210-t006:** The simulation results of different rainfall durations (r = 0.4).

Rainfall Duration (min)	Return Periods (a)	Cumulative Rainfall (mm)	Maximum Inundation Depth (m)	Inundation Extent (m^2^)	Proportion of Inundation Area (%)
60	1	40.3	0.803	78,115	4.13%
	2	45.2	0.864	99,016	5.24%
	5	51.6	1.093	134,200	7.11%
	10	56.4	1.199	181,229	9.59%
120	1	50.8	0.847	84,453	4.47%
	2	56.9	0.978	103,769	5.49%
	5	65.0	1.126	148,234	7.84%
	10	71.1	1.222	195,118	10.33%
180	1	57.3	0.857	87,236	4.62%
	2	64.2	0.979	106,455	5.63%
	5	73.3	1.135	155,823	8.25%
	10	80.2	1.229	201,027	10.64%

## Data Availability

The data presented in this study are available on request from the corresponding author.

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
