# Peer review of "Urban Inundation under Different Rainstorm Scenarios in Lin’an City, China"

_ijerph, 2022, doi:10.3390/ijerph19127210_

Round 1

Reviewer 1 Report

The manuscript "Urban Inundation under Different Rainstorm Scenarios in a Rapid Urbanization Region of China" is a presentation of the results obtained from the InfoWorks ICM model. The authors focused mainly on the results of the floods on the surface of the urban catchment area. They obtained confirmation that the model accurately reflects the water movement in the 2D system on the ground surface.

In my opinion, the manuscript lacks a section devoted to limiting sewer overflows during heavy rains. It is important not only to indicate the places where water flows from the sewer system, but also the most effective way to prevent these overflows. It would be good for the authors to supplement in a short description what technical elements of the transformation delay in the sewer system can be simulated in the model.

The numbering of figures should be corrected both under the figures and in the text.

The manuscript content documents the simulations carried out with the InfoWorks ICM model, which qualifies it as "average scientific novelty". There is no comparative analysis in which the obtained results would be confronted with the results from other models, eg MIKE. The discussion should include a section devoted to demonstrating that the InfoWorks ICM model works better than other models in a specific group of topics to be solved, and in some problems worse.

Author Response

Dear Reviewer,

The revised version of the manuscript (Manuscript ID: ijerph-1697954) has been submitted, which has been cautiously revised according to the reviewers’ comments.

We thank the reviewer for careful read and thoughtful comments on previous draft. We have carefully taken all comments into consideration in preparing our revision.

For details of the revision, please refer to the revised manuscript (with changes marked) and the following responses. Along with the manuscript, we provide our point-by-point response.

Response to Reviewer 1 Comments

Point 1: The manuscript "Urban Inundation under Different Rainstorm Scenarios in a Rapid Urbanization Region of China" is a presentation of the results obtained from the InfoWorks ICM model. The authors focused mainly on the results of the floods on the surface of the urban catchment area. They obtained confirmation that the model accurately reflects the water movement in the 2D system on the ground surface.

Response 1: We thank the reviewer very much for the positive comment.

Point 2: In my opinion, the manuscript lacks a section devoted to limiting sewer overflows during heavy rains. It is important not only to indicate the places where water flows from the sewer system, but also the most effective way to prevent these overflows. It would be good for the authors to supplement in a short description what technical elements of the transformation delay in the sewer system can be simulated in the model. 

Response 2:  Thank you for the suggestion. In lines 384-388 of the initial manuscript, we briefly put forward some measures to prevent overflows. As you suggested, we find these contents are not enough. According to the suggestion, we have expanded this section, please see the detailed revision in manuscript at line 397 to 406 (with changes marked).

Point 3: The numbering of figures should be corrected both under the figures and in the text.

Response 3: Thank you for the suggestion. The numbering of figures has been corrected both under the figures and in the text, and has been carefully checked.

Point 4: The manuscript content documents the simulations carried out with the InfoWorks ICM model, which qualifies it as "average scientific novelty". There is no comparative analysis in which the obtained results would be confronted with the results from other models, eg MIKE. The discussion should include a section devoted to demonstrating that the InfoWorks ICM model works better than other models in a specific group of topics to be solved, and in some problems worse. 

Response 4: Thank  you for the suggestion. We compared the simulation results of the InfoWorks ICM model with the SWMM model. The results show that the SWMM model can only simulate the overflow of the manholes, but not get the information of inundation depths and extents. Meanwhile, we compared the simulated flow results with the measured flow results, and the Nash–Sutcliffe efficiency coefficient is higher than 0.82.

Reviewer 2 Report

The manuscript deals with the impact of different rainstorm scenarios over urban inundation in a Region of China.

The manuscript fits well with the aim of the journal and it is well written. The work is well done, technical aspects are well described and addressed in the correct form.

Few minor revisions should be addressed to make the manuscript suited for publication:

In detail:

1)     In my opinion in the section “Introduction” few words should be addressed to the preferential flow process that occur in case of heavy rainstorms. It could not be neglected the impact on soil and subsoil of preferential pathways. At this regard, several scientific studies are increasingly referring this crucial question (see for example Nimmo,2020, De Carlo et al., 2021; Scaini et al,2018)

2)     Line 97 and Line 137. Error! Reference source not found. Please, correct the typo.

3)     How can soil moisture affect the inundation scenarios? Have the authors validated the inundation scenarios map with point scale measurements? Or incorporate point scale measurements into the model scenario?

4)     Can the inundation scenario maps be updated with other data into a data assimilation scheme? See for example Wang et al., 2021

References

Nimmo, J.R. The Processes of Preferential Flow in the Unsaturated Zone. Soil Sci. Soc. Am. J. 2020, doi:10.1002/saj2.20143.

De Carlo, L.; Perkins, K.; Caputo, M.C. Evidence of Preferential Flow Activation in the Vadose Zone via Geophysical Monitoring. Sensors 2021, 21, 1358. https://doi.org/10.3390/s21041358

Scaini, A.; Hissler, C.; Fenicia, F.; Juilleret, J.; Iffly, J.F.; Pfister, L.; Beven, K. Hillslope response to sprinkling and natural rainfall using velocity and celerity estimates in a slate-bedrock catchment. J. Hydrol. 2018, 558, 366–379, doi:10.1016/j.jhydrol.2017.12.011.

Wang, W.; Liu, J.; Li, C.; Liu, Y.; Yu, F. Data Assimilation for Rainfall-Runoff Prediction Based on Coupled Atmospheric-Hydrologic Systems with Variable Complexity. Remote Sens. 2021, 13, 595. https:// doi.org/10.3390/rs13040595

Author Response

Dear Reviewer,

The revised version of the manuscript (Manuscript ID: ijerph-1697954) has been submitted, which has been cautiously revised according to the reviewers’ comments.

We thank the reviewer for careful read and thoughtful comments on previous draft. We have carefully taken all comments into consideration in preparing our revision.

For details of the revision, please refer to the revised manuscript (with changes marked) and the following responses. Along with the manuscript, we provide our point-by-point response.

Response to Reviewer 2 Comments

Point 1: The manuscript deals with the impact of different rainstorm scenarios over urban inundation in a Region of China. The manuscript fits well with the aim of the journal and it is well written. The work is well done, technical aspects are well described and addressed in the correct form.

Response 1: We thank the reviewer very much for the positive comment.

Point 2: In my opinion in the section “Introduction” few words should be addressed to the preferential flow process that occur in case of heavy rainstorms. It could not be neglected the impact on soil and subsoil of preferential pathways. At this regard, several scientific studies are increasingly referring this crucial question (see for example Nimmo,2020, De Carlo et al., 2021; Scaini et al,2018) 

Response 2: Thank you for the suggestion. As suggested, we added the content of preferential flow process in the section “Introduction”. Please see the detailed revision in manuscript in the first paragraph of the “Introduction” (with changes marked).

Point 3: Line 97 and Line 137. Error! Reference source not found. Please, correct the typo.

Response 3: Thank you for the comments. It has been followed. Please see the detailed revision in manuscript at line 104 and line 145 (with changes marked).

Point 4: How can soil moisture affect the inundation scenarios? Have the authors validated the inundation scenarios map with point scale measurements? Or incorporate point scale measurements into the model scenario? 

Response 4: Thank you for the comments. Soil moisture will affect the simulation results. In the construction of the model, we use Horton's model to simulate the infiltration of precipitation into the soil, which is a commonly used runoff model in hydrological models. Horton model is a well-known infiltration formula, which assumes that the potential permeability decreases exponentially with time. However, this model does not account for initial moisture conditions adequately (Mallari, 2015). This is a deficiency in our research, and we will consider this problem in future research. We added this deficiency to the manuscript, please see the detailed revision in manuscript at line 422 to 426 (with changes marked).

As for point scale measurements, we validated the inundation scenarios map with point scale measurements. When the rainstorm came, we made an on-the-spot investigation and measured the inundation depth of the inundation area. Table 3 shows the comparison between the measured points and the simulation results.

Reference

Mallari, K.J.B.; Arguelles, A.C.C.; Kim, H.; Aksoy, H.; Kavvas, M.L.; Yoon, J. Comparative Analysis of Two Infiltration Models for Application in a Physically Based Overland Flow Model. Environ. Earth Sci. 2015, 74, doi:10.1007/s12665-015-4155-7.

Point 5: Can the inundation scenario maps be updated with other data into a data assimilation scheme? See for example Wang et al., 2021 

Response 5: Thank you for the comments. According to our current understanding, the inundation scenario maps can’t be one source of data assimilation. Wang et al. assimilated Global Telecommunications System (GTS) data with radar reflectivity data, and then put it into the hydrological model. However, the inundation scenario maps after assimilation can’t be input into the model for simulation. What we can do is to assimilate rainfall data from different sources, which requires further exploration.

References

Nimmo, J.R. The Processes of Preferential Flow in the Unsaturated Zone. Soil Sci. Soc. Am. J. 2020, doi:10.1002/saj2.20143.

De Carlo, L.; Perkins, K.; Caputo, M.C. Evidence of Preferential Flow Activation in the Vadose Zone via Geophysical Monitoring. Sensors 2021, 21, 1358. https://doi.org/10.3390/s21041358

Scaini, A.; Hissler, C.; Fenicia, F.; Juilleret, J.; Iffly, J.F.; Pfister, L.; Beven, K. Hillslope response to sprinkling and natural rainfall using velocity and celerity estimates in a slate-bedrock catchment. J. Hydrol. 2018, 558, 366–379, doi:10.1016/j.jhydrol.2017.12.011.

Wang, W.; Liu, J.; Li, C.; Liu, Y.; Yu, F. Data Assimilation for Rainfall-Runoff Prediction Based on Coupled Atmospheric-Hydrologic Systems with Variable Complexity. Remote Sens. 2021, 13, 595. https:// doi.org/10.3390/rs13040595

Reviewer 3 Report

The manuscript reports a well-developed and adequately described study. All necessary data are reported and adequately described and elaborated.

The manuscript  needs a thorough revision by an English-speaking expert.

Some comments to improve the quality of the manuscript are given directly on the pdf version of the manuscript 

Author Response

Dear Reviewer,

The revised version of the manuscript (Manuscript ID: ijerph-1697954) has been submitted, which has been cautiously revised according to the reviewers’ comments.

We thank the reviewer for careful read and thoughtful comments on previous draft. We have carefully taken all comments into consideration in preparing our revision.

For details of the revision, please refer to the revised manuscript (with changes marked) and the following responses. Along with the manuscript, we provide our point-by-point response.

Response to Reviewer 3 Comments

Point 1: The manuscript reports a well-developed and adequately described study. All necessary data are reported and adequately described and elaborated.

Response 1: We thank the reviewer very much for the positive comment.

Point 2: The manuscript  needs a thorough revision by an English-speaking expert. 

Response 2:  In this version, we use a professional English editing service by MDPI. The experts have carefully reviewed the manuscript and corrected the language mistake. 

Point 3: Line 97 and Line 137. Error! Reference source not found. Please, correct the typo.

Response 3: Thank you for the comments. It has been followed. Please see the detailed revision in manuscript at line 104 and line 145 (with changes marked).

Point 4: Some comments to improve the quality of the manuscript are given directly on the pdf version of the manuscript. 

Response 4: We considered all the comments mentioned in pdf in preparing our revision.
